# Generation of Conditional Knockout Alleles for PRUNE-1

**DOI:** 10.3390/cells12040524

**Published:** 2023-02-06

**Authors:** Xiaoli Wu, Louise R. Simard, Hao Ding

**Affiliations:** 1Department of Biochemistry and Medical Genetics, University of Manitoba, Winnipeg, MB R3E 0J9, Canada; 2Children’s Hospital Research Institute of Manitoba (CHRIM), Winnipeg, MB R3E 0Z3, Canada

**Keywords:** PRUNE1, CRISPR/Cas9, in vitro fertilization, zygote electroporation, conditional null allele, Cre recombinase, cerebellar Purkinje cells, neurodegeneration

## Abstract

PRUNE1 is a member of the aspartic acid-histidine-histidine (DHH) protein superfamily, which could display an exopolyphosphatase activity and interact with multiple cellular proteins involved in the cytoskeletal rearrangement. It is widely expressed during embryonic development and is essential for embryogenesis. PRUNE1 could also be critical for postnatal development of the nervous system as it was found to be mutated in patients with microcephaly, brain malformations, and neurodegeneration. To determine the cellular function of PRUNE1 during development and in disease, we have generated conditional mouse alleles of the *Prune1* in which *lox*P sites flank exon 6. Crossing these alleles with a ubiquitous Cre transgenic line resulted in a complete loss of PRUNE1 expression and embryonic defects identical to those previously described for *Prune1* null embryos. In addition, breeding these alleles with a Purkinje cell-specific Cre line (*Pcp2-Cre*) resulted in the loss of Purkinje cells similar to that observed in patients carrying a mutation with loss of PRUNE1 function. Therefore, the *Prune1* conditional mouse alleles generated in this study provide important genetic tools not only for dissecting the spatial and temporal roles of PRUNE1 during development but also for understanding the pathogenic role of PRUNE1 dysfunction in neurodegenerative or neurodevelopmental disease. In addition, from this work, we have described an approach that allows one to efficiently generate conditional mouse alleles based on mouse zygote electroporation.

## 1. Introduction

PRUNE1 is homologous to yeast polyphosphatase PPX1, containing the aspartic acid-histidine-histidine (DHH) motif (at the N-terminus) and an exopolyphosphatase activity for hydrolyzing short-chain polyphosphates [1,2]. It could also display phosphodiesterase activity to hydrolyze the second messenger cyclic adenosine monophosphate (cAMP) [3]. Besides its enzymatic activities, PRUNE1 also contains the motifs (at the C-terminus) responsible for interacting with several intracellular proteins mainly involved in cytoskeletal rearrangement, including nucleoside-diphosphate kinase A/B (or Nm23-H1/H2), glycogen synthase kinase 3β (GSK-3β), and α- and β-tubulins [4,5,6]. These interactions could play important roles in cellular division and migration [7].

PRUNE1 has also been demonstrated to be essential for mouse embryonic development. *Prune1^−/−^* mice were embryonic lethal and developed progressive abnormalities in the heart and vasculature, implicating its role in angiogenesis [2]. However, how PRUNE1 contributes to embryonic development is currently unknown. PRUNE1 could also be required for postnatal development of the nervous system as indicated by human genetic studies, which led to the identification of several homozygous and compound heterozygous mutations of this gene in patients with microcephaly, brain malformations, and neurodegeneration [2,6,8,9,10]. Particularly, a homozygous loss of function mutation of *PRUNE1* (i.e., c.521-2A > G in a canonical splice site) was found to be associated with fatal human neurodegeneration characterized by ataxia, hypotonia, contractures, and early childhood death [9,11]. Pathologically, patients with this mutation showed cerebellar hypoplasia with massive loss of Purkinje cells and other neurons [9], supporting the critical role of PRUNE1 in neuronal survival during postnatal development.

To determine the role of PRUNE1 during development and in disease, in this study, we generated a *Prune1* conditional mouse allele that allows to inactivate PRUNE1 through Cre-mediated recombination in a cell-specific manner and also to circumvent the embryonic lethality of the *Prune1* null allele.

## 2. Materials and Methods

### 2.1. Generation of Prune1 Conditional Mouse Alleles

A CRISPR/Cas9-based gene targeting method was used to sequentially introduce *lox*P sites on the 5′ and 3′ sides of exon 6 of mouse *Prune1*. Briefly, guide RNAs (gRNA1 with an on-target score of 77: 5′-tctttgagcacgactactaa-3′ to target a 3′ *lox*P in intron 6, and gRNA2 with an on-target score of 78: 5′-tacaaagtccctcaatctac-3′ to target a 5′ *lox*P in intron 5) were synthesized by Integrated DNA Technologies (IDT; Coralville, IA, USA). The on-target scores of these gRNAs were calculated based on IDT software (Predesigned Alt-R^TM^ CRISPR-Cas9 guide RNA). Donor DNAs containing the *lox*P sequence used in this study (Appendix A) were also synthesized by IDT. These genetic components, together with Cas9 (Alt-R^®^ S.p. Cas9 Nuclease V3, IDT, Cat# 1081061), were electroporated into C57BL/6J zygotes generated by in vitro fertilization as previously described [12,13,14]. Electroporation was done using the BioRad Gene Pulser Xcell^TM^ (Bio-Rad, Mississauga, ON, Canada) at 30 V, 1 s ON, 99 s OFF, for 12 cycles. After electroporation, zygotes were either cultured to the blastocyst stage for PCR analysis to test targeting efficiency or to the 2-cell stage and then transferred to CD1 pseudopregnant mice (0.5 dpc) for generating offspring.

Electroporation of 100 ng/µL of gRNA, 200 ng/µL of donor DNA and 50 ng/μL Cas9 in a 10 µL volume of Opti-MEM^®^ (Thermo Fisher Scientific,Mississauga, ON, Canada, Cat# 11058021) was used to generate mice homozygous for 3′ *lox*P. The sperm of these mice were collected for in vitro fertilization of C57BL/6J oocytes to produce zygotes that were used for the second round of targeting for 5′ *lox*P. These two consecutive rounds of gene targeting resulted in multiple mouse lines, which contained homozygous 5′ *lox*P and heterozygous 3′ *lox*P. These mouse lines were bred with wt C57BL/6J mice for two generations to segregate the potential off-targets. The resultant mice were intercrossed to generate a homozygous *Prune1* conditional allele (*Prune1^F/F^*). We used DNA sequencing to confirm the corrected targeted *lox*P sites in this mouse allele.

### 2.2. Characterization of the Prune1 Conditional Mouse Allele

*Prune1^F/F^* mice were bred with the *EIIa-Cre* line (The Jackson Laboratory, Bar Harbor, ME, USA, stock #003724), which displays a robust Cre activity in germ cells [15], to generate the *Prune1^Δexon6^* mice. These mice were bred to homozygotes to determine whether *Prune1^F/F^* can function as a null after Cre-mediated excision.

*Prune1^F/F^* mice were also bred with *Pcp2-Cre* mice [16] (The Jackson Laboratory, stock #010536) to generate *Prune1^F/F^/Pcp2-Cre* mice to conditionally knock out PRUNE1 in cerebellar Purkinje cells and determine whether *Prune1^F/F^* mice can be used to model human neurodegeneration associated with loss of PRUNE1 function.

### 2.3. Genotyping

PCR was applied to genotype mice using ear-punched DNA. PCR was also used to genotype blastocysts using the Quanta Accustart II Mouse Genotyping kit (Quantabio, Beverly, MA, USA, Cat#95135-500). To genotype the *Prune1* conditional allele, sense primer (5′-ggctagcctagaactcatgttg-3′) and antisense primer (5′-gacatacatgcaggcaaaacac-3′) were designed to detect the wt (249 bp) and conditional (284 bp) allele, respectively. PCR products were detected by polyacrylamide gel electrophoresis. To genotype the *Prune1^Δexon6^
*allele, sense primer (5′-ggctagcctagaactcatgttg-3′) and antisense primer (5′-ctttgccactttgtagtggttc-3′) were used to detect the mutant (329 bp) and the wt (1478 bp) alleles, respectively. To genotype the *Pcp2-Cre* and *EIIa-Cre* alleles, PCR was done based on information provided by the Jackson Laboratory.

### 2.4. Histology and Immunohistochemistry (IHC)

Mouse embryos were dissected and fixed with 10% formalin. Adult *Prune1^F/F^/Pcp2-Cre* mouse brains were collected after transcardial perfusion with phosphate-buffered saline (PBS) followed by 4% paraformaldehyde (PFA) and were further fixed with 4% PFA overnight. Five μm paraffin-embedded tissue sections were used for histology and IHC. Histology was done by staining either with Hematoxylin-eosin (H&E) or Alexa Fluor488 conjugated wheat germ agglutinin (WGA) (Invitrogen, Burlington, ON, Canada, Cat#11261), which labels cell membrane.

IHC was performed as described previously [17,18]. Briefly, sections were pretreated with Retrieval solution (Dako, Burlington, ON, Canada) and blocked with either mouse IgG blocking reagent (Vector Laboratory, Newark, CA, USA) or serum-free blocking reagent (Dako), and then incubated with primary antibodies overnight at 4°. Primary antibodies used were rat anti-Endomucin (dilution 1:1000, Abcam, Cambridge, UK, ab106100), rabbit anti-E-cadherin (dilution 1:1000, Abcam ab53033), and rabbit anti-Calbindin (1:1000, MilliporeSigma, Oakville, ON, Canada, AB1778) antibodies. Secondary antibodies included Alexa Fluor488- or Fluor568 conjugated goat anti-rabbit or rat (Thermo Fisher Scientific) and biotinylated anti-rabbit antibody. Signals were detected either by immunofluorescence or staining with metal 3,3′-diaminobenzidine tetrachloride. Immunofluorescence images with *DAPI* (4′,6-diamidino-2-phenylindole) as counterstaining were collected using a Zeiss Axioplan 2 microscope to generate Z-stacks, which were processed using extended focus in Axio Vision 4.6.

### 2.5. Western Blot Analysis

Protein extracts from MEF cells were prepared with ice-cold PBS buffer containing 1× Halt Protease and Phosphatase Inhibitor Cocktail (Thermo Fisher Scientific, Cat#78440) and 1% Triton X-100. Fifty μg lysate was separated by 8% SDS-PAGE and transferred to a nitrocellulose membrane, which was blocked with 5% de-fatted milk and incubated with mouse anti-PRUNE1 antibody (dilution 1:500, Abcam ab88613). Protein was detected using the enhanced chemiluminescence (Thermo Fisher Scientific, Cat#A38556) and imaged with Azure 400 imaging system.

## 3. Results and Discussion

The mouse *Prune1* gene consists of eight exons, encoding PRUNE1 protein comprised of 454 amino acids, which share 85% identity or 90% similarity with human PRUNE1. No other isoforms of PRUNE1 could be expressed from this locus [19]. In order to establish the conditional mouse alleles that allow completely knocking out the PRUNE1 activity through Cre-mediated excision, we decided to target exon 6 of the *Prune1* gene by flanking it with two *lox*P sites (Figure 1a). Deletion of this exon is predicted to generate a null allele as it can cause a frame-shift mutation after splicing of exons 5 and 7, resulting in a truncated mRNA containing multiple premature termination codons, which are known to be degraded via nonsense-mediated mRNA decay [20].

We applied a CRISPR/Cas9-based gene-targeting method to sequentially introduce two *lox*P sites on the 5′ and 3′ sides of exon 6 of mouse *Prune1* (Figure 1b). To improve the efficiency of this procedure, firstly, we used an in vitro assay to test the targeting efficiency of the CRISPR/Cas9 components used in this study. In this assay, wild-type (wt) C57BL/6J zygotes generated by in vitro fertilization were electroporated with different amounts of gRNA, donor DNA, and Cas9 protein, and then cultured to blastocytes for PCR-based genetic analysis. From this in vitro analysis, we found that: (1) Zygote electroporation of 100 ng/µL of gRNA, 200 ng/µL of donor DNA and 50 ng/μL Cas9 in a 10 µL volume of Opti-MEM^®^ could allow targeting each individual *lox*P site as homozygotes in the *Prune1* locus without significantly affecting the survival of electroporated zygotes, and (2) gRNA/donor DNA used for targeting 5′ *lox*P showed significantly higher efficiency of generating homozygotes than the one used for targeting 3′ *lox*P (Appendix A). Based on this finding, we decided to perform the first round of targeting to generate mice homozygous for 3′ *lox*P. This is because using this mouse allele to do the second round of targeting for the 5′ *lox*P should have a higher frequency for generating mice with both 5′ *lox*P and 3′ *lox*P sites on the same DNA strand, such as mice containing homozygous 5′ *lox*P and heterozygous 3′ *lox*P.

From the first round of targeting, we obtained four mice (two males and two females) homozygous for 3′ *lox*P out of ~150 electroporated zygotes generated by in vitro fertilization. The sperm of these mice were harvested for in vitro fertilization to produce zygotes that were used for the second round of 5′ *lox*P targeting. This procedure led to efficiently establishing multiple mouse lines that contained homozygous 5′ *lox*P and heterozygous 3′ *lox*P (Figure 1c). We bred these mouse lines with wt C57BL/6J mice for two generations to segregate the potential off-targets. The resultant mice were intercrossed to generate homozygous *Prune1* conditional allele (*Prune1^F/F^*). We used DNA sequencing to confirm the corrected targeted *lox*P sites in this mouse allele (Appendix A). *Prune1^F/F^* mice are postnatal viable, fertile, and behave as wt mice. Two independent *Prune1^F/F^* mouse lines (#1 and #6 indicated in Figure 1c) were used for further characterization.

To determine in vivo excision of the *Prune1^F/F^* allele and also to verify that it can function as a null after Cre-mediated excision, we bred these mice with the *EIIa-Cre* mice, which harbor a robust Cre activity in germ cells [15], to delete the floxed exon 6, generating the *Prune1^Δexon6^* mutants (Appendix A). To demonstrate that the *Prune1^Δexon6^
*allele is truly a null allele, we performed heterozygous crosses to produce homozygous *Prune1^Δexon6/^^Δexon6^* embryos. The complete absence of PRUNE1 protein in these mutant embryos was confirmed by Western blot analysis (Figure 2a). *Prune1^Δexon6/^^Δexon6^* embryos died between embryonic day 10.5 (E10.5) and E12.5. These mutants displayed marked cardiac hypoplasia with pericardial effusion and a thin myocardial wall (Figure 2c,g), as well as significant vascular remodeling defects in the yolk sac and embryo proper, such as an avascular yolk sac with primitive vascular plexus (Figure 2b,e) and lack of invaded vascular plexus in the neural tube (Figure 2i). All these developmental defects are similar to the previously published phenotypes for *Prune1^−/−^* embryos [2], indicating a full disruption of PRUNE1 function in the *Prune1^Δexon6/^^Δexon6^* mutant embryos. Consistent with the widespread expression of PRUNE1 during mouse embryogenesis [2,21], we found that *Prune1^Δexon6/^^Δexon6^* embryos also displayed severe defects in hematopoiesis and in the differentiation of trophoblasts (Figure 2e and Figure 3). Therefore, loss of PRUNE1 function likely affects multiple cell lineages during embryonic development. Future conditional knockout studies with our established *Prune1^F/F^* mice should enable us to dissect the role of PRUNE1 in these cell lineages to better understand its role during embryonic development.

Finally, we used the *Prune1^F/F^* allele to determine whether PRUNE1 is essential for the survival of cerebellar Purkinje cells, as indicated by human patients carrying a mutation with loss of PRUNE1 function (i.e., c.521-2A > G in a canonical splice site) [9,11]. For this, we bred the *Prune1^F/F^* allele with *Pcp2-Cre* mice to generate *Prune1^F/F^/Pcp2-Cre* mice to specifically knock out PRUNE1 in Purkinje cells. *Pcp2-Cre* mice have been shown to display highly restricted Cre activity in Purkinje cells, starting at postnatal day 6 (P6) and reaching a plateau around P15 [16]. Therefore, we applied histological approaches to characterize *Prune1^F/F^/Pcp2-Cre* mice after P15. We found that these mice started to show Purkinje cells degeneration at P20 and lost more than 90% of these cells by the age of two months old (Figure 4). This defect fully recapitulates the pathological hallmark of human patients. Therefore, future studies on using the *Prune1^F/F^* allele to determine the cell-autonomous role of PRUNE1 in these cells will greatly help to understand the pathogenesis of this disease mutation.

In conclusion, we have generated the first conditional allele for *Prune1*, allowing Cre-mediated inactivation of PRUNE1 function during mouse development and in specific adult tissues. This allele will be an important tool to dissect the developmental and cellular function of PRUNE1 in a tissue- or cell-specific manner and to better understand the pathogenic role of PRUNE1 mutations in human neurodegenerative or neurodevelopmental disorders.

Furthermore, from this work, we have also described a zygote electroporation-based approach that allows one to efficiently generate mouse conditional alleles. Due to its simplicity, zygote electroporation is still widely applied for creating a conditional allele by sequentially targeting two *lox*P sites in the mouse genome. However, the challenge of this method is how to efficiently target two *lox*P sites on the same DNA strand. As described in this study, we used a simple in vitro assay to first identify the conditions that allow us to use CRISPR/Cas9 to target *lox*P sites as homozygotes in the mouse genome, which can solve this problem and significantly improve the efficiency of generating conditional alleles. Based on this study, as well as our other gene-targeting experiments, gRNAs with on-target scores above 60 have the potential for targeting *lox*P sequence or small DNA fragments (such as a coding sequence of protein tag) as homozygotes in the mouse genome, which can be optimized using the in vitro assay as described in this study.

## Figures and Tables

**Figure 1 cells-12-00524-f001:**
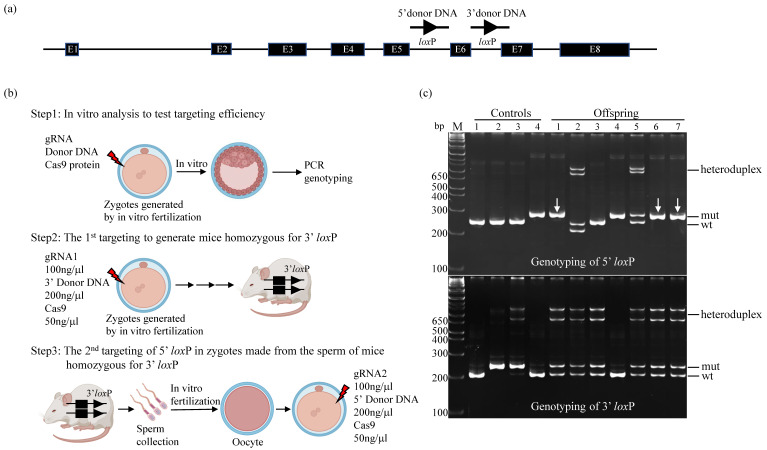
Generation of the *Prune1* conditional allele by CRISPR/Cas9-based gene-targeting. (**a**) Genomic structure of the mouse *Prune1* locus. The regions used for inserting two *lox*P sites that flank exon 6 (E6) are shown. (**b**) Schematic representation of a targeting procedure that allowed to efficiently generate the *Prune1* conditional alleles within five months. It consists of three steps: (1) In vitro analysis to test targeting efficiency, (2) the first round of targeting to generate mice homozygous for 3′ *lox*P, and (3) the second round of targeting of the 5′ *lox*P in zygotes made from the sperm of mice homozygous for 3′ *lox*P. (**c**) PCR genotyping of mouse offspring generated from the second round of targeting. Among seven offspring, three contained homozygous 5′ *lox*P and heterozygous 3′ *lox*P (1, 6, and 7 indicated by arrows), which was further confirmed by DNA sequencing. The PCR products of mutant (mut) and wt, as well as the formation of heteroduplexes (which reflects heterozygosity), are indicated. Controls used in this genotyping were genomic DNA prepared from: (1) wt C57BL/6J; (2) and (3): Blastocysts carrying homozygous 3′ *lox*P, and (4) blastocyst carrying homozygous 5′ *lox*P. The base pair (bp) of DNA marker (M) is indicated.

**Figure 2 cells-12-00524-f002:**
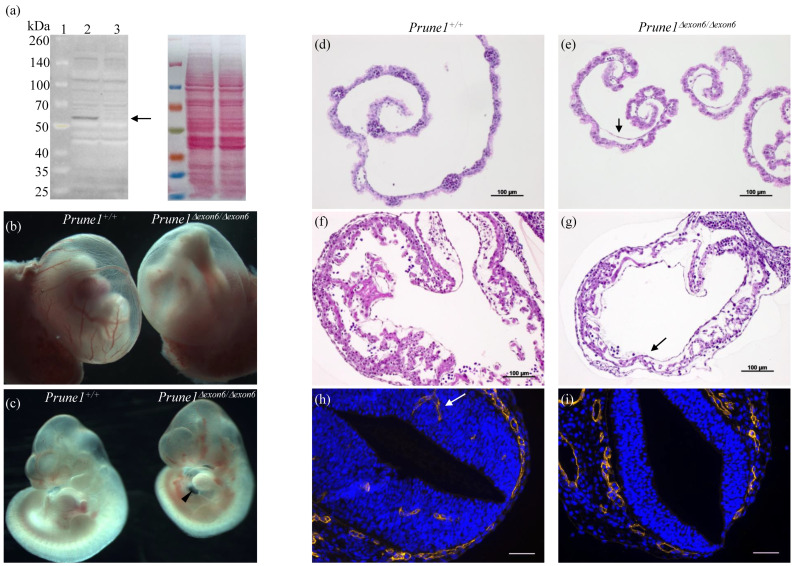
Characterization of the *Prune1^Δexon6/^^Δexon6^* as a null allele. (**a**) Western blot analysis, demonstrating the complete absence of PRUNE1 protein in mouse embryonic fibroblast (MEF) cells derived from E10.5 *Prune1^Δexon6/^^Δexon6^* embryo. Lane 1, Protein molecular weight marker; Lane 2, wt MEF; Lane 3, the *Prune1^Δexon6/^^Δexon6^* MEF. The arrow indicates PRUNE1 protein (~58 kDa) which was only detected in wt MEF cells. Ponceau S staining is shown as a normalization control. (**b**,**c**) Whole mount imaging of yolk sac and embryos dissected from *Prune1^Δexon6^* intercross at E10.5. The *Prune1^Δexon6/^^Δexon6^* displayed an avascular yolk sac with primitive vascular plexus and pericardial effusion (arrowhead) compared to the control littermate. (**d**–**g**) Hematoxylin-eosin (H&E) staining of sections prepared from E10.5 yolk sac and embryos showing distended capillary with defective hematopoiesis (arrow in (**e**)) and a thin heart wall with severe reduction of cardiomyocytes and very little trabeculae (arrow in (**g**)) in the *Prune1^Δexon6/^^Δexon6^* mutants compared to the control. (**h**,**i**) Immunostaining of transverse sections of E10.5 embryos with anti-Endomucin antibody (a marker of endothelial cells), demonstrating lack of invaded vascular plexus in the neural tube of *Prune1^Δexon6/^^Δexon6^* compared to the control. Arrow indicates invaded blood vessel in the control neural tube.

**Figure 3 cells-12-00524-f003:**
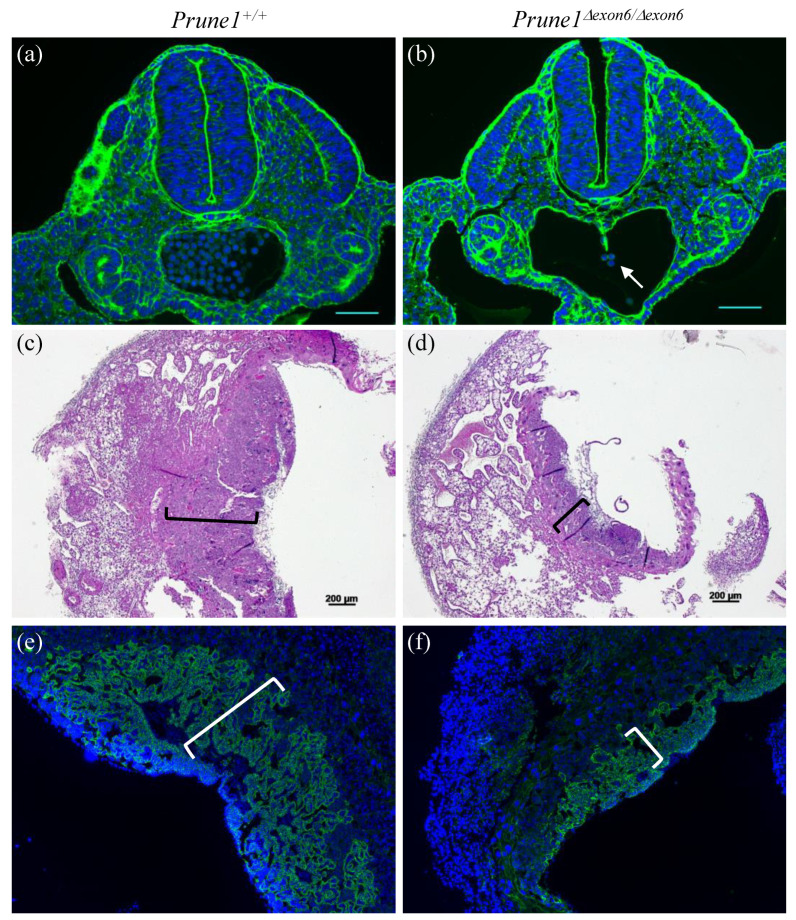
Additional developmental defects presented in the *Prune1^Δexon6/^^Δexon6^* mutants. (**a**,**b**) Wheat germ agglutinin (WGA) staining of transverse sections of E10.5 embryos showing significantly decreased hematopoietic cells in the aorta-gonad-mesonephros (AGM) of *Prune1^Δexon6/^^Δexon6^* (arrow in (**b**)) compared to the control. (**c**,**d**) H&E staining of E10.5 placenta, indicating a significantly thinner layer of trophoblast cells in the chorionic labyrinth (marked by brackets) in the *Prune1^Δexon6/^^Δexon6^* placenta compared to the wt control. (**e**,**f**) Immunostaining of E10.5 placenta with anti-E-cadherin antibody (a global marker of syncytiotrophoblast), demonstrating significantly decreased labyrinthine syncytiotrophoblasts (marked by brackets) in the *Prune1^Δexon6/^^Δexon6^* placenta compared to the wt control.

**Figure 4 cells-12-00524-f004:**
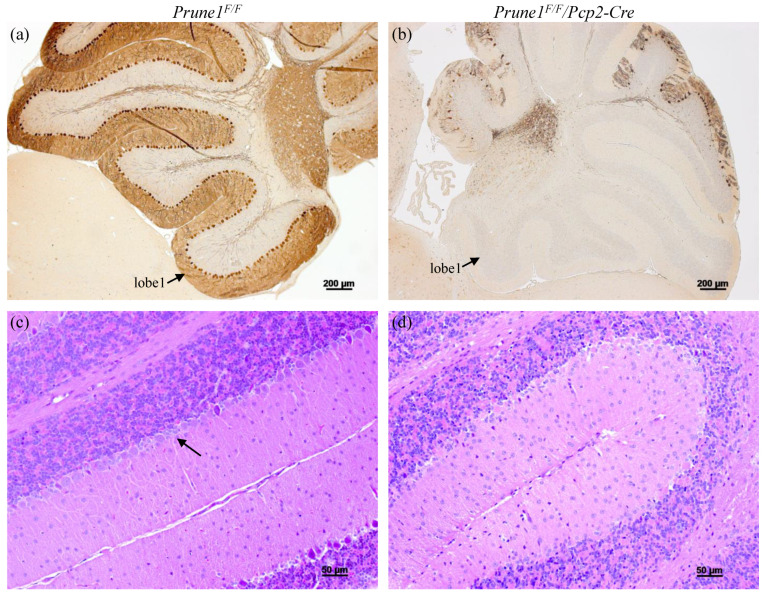
Characterization of the *Prune1^F/F^/Pcp2-Cre* mice with conditional knockout of PRUNE1 in cerebellar Purkinje cells. (**a**,**b**) Immunostaining with anti-Calbindin antibody (a marker for Purkinje cell), demonstrating loss of more than 90% of Purkinje cells in two months old *Prune1^F/F^/Pcp2-Cre* cerebella. Lobe 1 of the cerebellum is indicated. (**c**,**d**) H&E staining further confirming the loss of Purkinje cells in two months old *Prune1^F/F^/Pcp2-Cre* cerebella. Arrow indicates Purkinje cell layer, which was present in the control, but not in the *Prune1^F/F^/Pcp2-Cre* cerebella.

## Data Availability

The data that support the findings of this study are available from the corresponding author upon reasonable request.

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
