# Peer review of "Generation of Conditional Knockout Alleles for PRUNE-1"

_cells, 2023, doi:10.3390/cells12040524_

Round 1

Reviewer 1 Report

PRUNE1 was found to be mutated in patients with microcephaly, brain malformations and neurodegeneration. However, the cellular function of PRUNE1 during development and in disease has been difficult to analyze in Prune1 KO mice due to embryonic lethality. In this study, the authors generated and analyzed Prune1-conditional KO mice, with a particular focus on neurodegenerative or neurodevelopmental disease. The result of this experiment is valuable and interesting; however, I think that there are several minor improvements that should be made before publication.

1)    L62-L63: Could you explain the meaning of the “on-target score” and how the scores were calculated (e.g., software or web site used, etc.)?

2)    L64-64: DNA sequence of donor DNA including loxP and homology arm should be shown.

3)    L157-L169: The reviewer is intrigued and surprised by the high efficiency of homozygous KI of loxP (3 out of 7 offspring contain homozygous 5’ loxP allele?). Generally, it may be difficult to generate homozygous loxP KI allele by 1st round genome editing. In many cases, heterozygous alleles including a loxP KI allele and an indel mutation or wild type allele are generated. Could you describe more detailed data of KI efficiency of in vitro analysis using blastocysts (Fig.1b, Step1) and the derivative offspring?

4)    DNA sequence around the two loxPs of established floxed mice is required.

5)    Fig.2: Western blot analysis confirmed the complete loss of PRUNE1 protein; however, genotyping PCR data to detect mutant allele (329 bp) and wt (1478 bp) should be shown.

6)    L217-L219: The authors described that Prune1F/F /Pcp2 Cre mice can be used to human disease with PRUNE1 dysfunction. However, the reviewer believes that mice with the c.521-2A>G mutation in PRUNE1 are the best model for human disease. Is it difficult to establish Prune1 base substitution mice due to embryonic lethality?

Author Response

Thanks a lot for your quick reviewing on our manuscript. Please see attached my response to your comments.

Best,

Hao Ding Ph.D

Reviewer 2 Report

The authors have generated conditional alleles for Prune1, allowing Cre-mediated inactivation of PRUNE1 function during mouse development and in Purkinje cells in the adult. Interesting as the topic is, I have some concerns the authors should address before the manuscript is acceptable for publication.

The authors use a zygote electroporation based approach. Cell survival rate, and electroporation insertion rate should be specified in the Material and Methods section. Also, the authors should describe how did they evaluate the off target effects in the generation of Prune1 conditional mouse alleles using the CRISPR/Cas9 based gene targeting method.

Removing exon 6 is not the same as a single nucleotide substitution (c.521-2A>G mutation in humans). Hence, I am concerned that the strategy the authors used is not appropriate to model this specific mutation-derived human disease, even if it has been shown to generate loss of function (LOF).

According to the authors “This defect fully recapitulates the pathological hallmark of patients with c.521-2A>G mutation of PRUNE1, demonstrating that the Prune1F/F allele will be valuable for modelling human diseases with loss of PRUNE1 function (lines 227-229). I strongly disagree with this statement. If the point mutation generates LOF, the authors can claim that their model recapitulates the LOF described for that mutation. However, the biological relevance of that mutation may go beyond the already observed LOF. The authors should discuss this, and moderate their assumptions. The authors could have well used the CRISPR/Cas9 system to introduce that point mutation and generate a model truly resembling that found in humans.

Author Response

Thanks a lot for your quick reviewing on our manuscript. Please see attached my response to your comments.

Best,

Hao Ding, Ph.D

Reviewer 3 Report

The manuscript of Wu and colleagues entitled “Generation of conditional knockout alleles for PRUNE1” reports the generation of conditional mouse allele trough insertion of two LoxP sites besides Prune1 exon 6. By crossing these alleles with a ubiquitous Cre transgenic line resulted in a complete loss of PRUNE1 expression and embryonic defects recapitulating those previously described for the Prune1 null embryos. Moreover, crossing these alleles with Purkinje cell-specific Cre line 16 (Pcp2-Cre) resulted in loss of Purkinje cells as observed in patients carrying a mutation with loss of PRUNE1 function. Overall, the authors described an efficient approach to generate mouse conditional alleles based on mouse zygote electroporation and generated conditional Prune1 exon 6 mice.

The manuscript is well conducted, with no major concerns. I have only some minor points:

-          The title is too vague and could results, for some reader, somehow misleading. In fact, there is no indication of the targeted exon. I suggest to edit the title to include this information

-          It is no clear why authors throughout the manuscript report the generation of multiple “alleles” while they are generating a single knockout allele (delta6).

-          In figure 1, the size of the bands as well as the marker line are missing.

-          In the results and discussion section, I will put more emphasis on the exploited approach to generate the conditional prune allele.

-          Authors should discuss in more detail which is the difference, in term of timing and molecular aspects, between the prune1 null embryos and the knockout embryos generated by crossing the conditional prune mice with the Cre-transgenic line. Shouldn’t these mice be somehow identical? The clear advantage of this mouse strain is the possibility to evaluate prune1 loss trough exploitation of inducible, or tissue specific, Cre expression.

Author Response

(The authors gave the same response as above.)

Round 2

Reviewer 2 Report

The authors have replied to my concerns. They couldn't address all of them, in order not to affect a future publication on PRUNE1 knock-in mouse study. However, they moderated their conclusions and inferences, and I think the manuscript is overall better.